# Learning How to Ground a Plan – Partial Grounding in Classical Planning

**Daniel Gnad, Álvaro Torralba**
Saarland University
Saarland Informatics Campus
Saarbrücken, Germany
{gnad,torralba}@cs.uni-saarland.de

**Martín Domínguez, Carlos Areces, Facundo Bustos**
Universidad Nacional de Córdoba
Córdoba, Argentina
{mardom75, carlos.areces, facundojosebustos}@gmail.com

## Abstract

Current classical planners are very successful in finding (non-optimal) plans, even for large planning instances. To do so, most planners rely on a preprocessing stage that computes a grounded representation of the task. Whenever the grounded task is too big to be generated (i.e., whenever this preprocess fails) the instance cannot even be tackled by the actual planner. To address this issue, we introduce a partial grounding approach that grounds only a projection of the task, when complete grounding is not feasible. We propose a guiding mechanism that, for a given domain, identifies the parts of a task that are relevant to find a plan by using off-the-shelf machine learning methods. Our empirical evaluation attests that the approach is capable of solving planning instances that are too big to be fully grounded.

## Introduction

Given a model of the environment, classical planning attempts to find a sequence of actions that lead from an initial state to a state that satisfies a set of goals. Planning models are typically described in the Planning Domain Definition Language (PDDL) (McDermott et al. 1998) in terms of predicates and action schemas with arguments that can be instantiated with a set of objects. However, most planners work on a grounded representation without free variables, like STRIPS (Fikes and Nilsson 1971) or FDR (Bäckström and Nebel 1995). Grounding is the process of translating a task in the lifted (PDDL) representation to a grounded representation. It requires to compute all valid instantiations that assign objects to the arguments of predicates and action parameters, even though only a small fraction of these instantiations might be necessary to solve the task.

The size of the grounded task is exponential in the number of arguments in predicates and action schemas. Although this is constant for all tasks of a given domain, and grounding can be done in polynomial time, it may still be prohibitive when the number of objects is large and/or some predicates or actions have many parameters.

The success of planners like FF (Hoffmann and Nebel 2001a) or LAMA (Richter, Westphal, and Helmert 2011) in finding plans for large planning tasks is undeniable. However, since most planners rely on grounding for solving a task, they fail without even starting the search for a plan whenever an instance cannot be grounded, making grounding a bottleneck for the success of satisficing planners.

Grounding is particularly challenging in open multi-task environments, where the planning task is automatically generated with all available objects even if only a few of them are relevant to achieve the goals. For example, in robotics, the planning task may contain all objects with which the robot may interact even if they are not needed (Lang and Toussaint 2009). In network-security environments, like the one modeled in the Caldera domain (Miller et al. 2018), the planning task may contain all details about the network. However, to the best of our knowledge, no method exists that attempts to focus the grounding on relevant parts of the task.

We propose *partial grounding*, where, instead of instantiating the full planning task, we focus on the parts that are required to find a plan. The approach is sound – if a plan is found for the partially grounded task then it is a valid plan for the original task – but incomplete – the partially grounded task will only be solvable if the operators in at least one plan have been grounded. To do so, we give priority to operators that we deem more relevant to achieve the goal. Inspired by relational learning approaches to domain control knowledge (e.g., Yoon, Fern, and Givan (2008), de la Rosa et al. (2011), Krajnansky et al. (2014)), we use machine learning methods to predict the probability that a given operator belongs to a plan. We learn from small training instances, and generalize to larger ones by using relational features in standard classification and regression algorithms (e.g., Kramer, Lavrač, and Flach (2001)). As an alternative model, we also experiment with relational trees to learn the probabilities (Muggleton and Raedt 1994).

Empirical results show that our learning models can predict which operators are relevant with high accuracy in several domains, leading to a very strong reduction of task size when grounding and solving huge tasks.

## Background

Throughout the paper, we assume for simplicity that tasks are specified in the STRIPS subset of PDDL (Fikes and Nilsson 1971). Our algorithms and implementation, however, are directly applicable to a larger subset of PDDL containing ADL expressions (Pednault 1989).

A lifted (PDDL) task $\Pi^{PDDL}$ is a tuple $(\mathcal{P}, \mathcal{A}, \Sigma^C, \Sigma^O, I, G)$ where $\mathcal{P}$ is a set of (first-order) atomic

*predicates*, $\mathcal{A}$ is a set of *action schemas*, $\Sigma := \Sigma^C \cup \Sigma^O$ is a non-empty set of *objects* consisting of constants $\Sigma^C$, and non-constant objects $\Sigma^O$, $I$ is the *initial state*, and $G$ is the *goal*. Predicates and action schemas have parameters. We denote individual parameters with $x, y, z$ and sets of parameters with $X, Y, Z$. An action schema $a[X]$ is a triple $(\mathsf{pre}(a), \mathsf{add}(a), \mathsf{del}(a))$, consisting of *preconditions*, *add list*, and *delete list*, all of which are subsets of $\mathcal{P}$, possibly pre-instantiated with objects from $\Sigma^C$, such that $X$ is the set of variables that appear in $\mathsf{pre}(a) \cup \mathsf{add}(a) \cup \mathsf{del}(a)$. $I$ and $G$ are subsets of $\mathcal{P}$, instantiated with objects from $\Sigma$.

A lifted task $\Pi^{PDDL}$ can be divided into two parts: the domain specification $(\mathcal{P}, \mathcal{A}, \Sigma^C)$ which is common to all instances of the domain, and the problem specification $(\Sigma^O, I, G)$ which is different for each instance of a domain.

A STRIPS task $\Pi$ is a tuple $(F, O, I, G)$, where $F$ is a set of grounded predicates, called *facts*, and $O$ is a set of grounded action schemas, called *operators*. A *state* $s \subseteq F$ is a set of facts, $I \subseteq F$ is the *initial state* and $G \subseteq F$ is the *goal*. An operator $o$ is *applicable* in a state $s$ if $\mathsf{pre}(o) \subseteq s$. In that case, the outcome state is $s' = (s \setminus \mathsf{del}(o)) \cup \mathsf{add}(o)$, and we write $s \xrightarrow{o} s'$ for the transition from $s$ to $s'$ via $o$. For a sequence of operators $\bar{o}$, we write $s \xrightarrow{\bar{o}} t$ if the operators in $\bar{o}$ can be iteratively applied to $s$, resulting in $t$. A sequence $\bar{o}$, with $I \xrightarrow{\bar{o}} s_G$, is a *plan* for $\Pi$ if $G \subseteq s_G$. A task $\Pi$ is solvable if a plan exists. The plan is *optimal* if its length is minimal among all plans for $\Pi$.

We define the *delete-relaxation* of a task $\Pi$ as the task $\Pi^+$ obtained by setting $\mathsf{del}(o) = \emptyset$, for all $o \in O$. We say that $\Pi$ is *delete-relaxed solvable* if $\Pi^+$ is solvable.

Given a lifted task $\Pi^{PDDL}$, we can compute the corresponding STRIPS task $\Pi$ by *instantiating* the predicates and action schemas with the objects in $\Sigma$. Then, $F$ contains a fact for each possible assignment of objects in $\Sigma$ to the arguments of each predicate $P[X] \in \mathcal{P}$, and $O$ contains an operator for each possible assignment of objects in $\Sigma$ to each action schema $a[X] \in \mathcal{A}$. In practice, we do not enumerate all possible assignments of objects in $\Sigma$ to the arguments in facts and action schemas. Instead, only those facts and operators are instantiated that are delete-relaxed reachable from the initial state (Helmert 2009).

## Partial Grounding

We base our method on the grounding algorithm of Fast Downward (Helmert 2006). To ground a planning task, this algorithm performs a fix-point computation similar to the computation of relaxed planning graphs (Blum and Furst 1997), where a queue is initialized with the facts in the initial state and in each iteration one element of the queue is popped and processed. If the element is a fact, then those operators of which all preconditions have already been processed (are reached) are added to the queue. If the element is an operator, all its add effects are pushed to the queue. The algorithm terminates when the queue is empty. Then, all processed facts and operators are delete-relaxed reachable from the initial state. For simplicity, the algorithm we describe here considers only STRIPS tasks but it can be adapted to support other PDDL features like negative preconditions or

---

**Algorithm 1:** Partial Grounding.

**Input:** A lifted task $\Pi^{PDDL} = (\mathcal{P}, \mathcal{A}, \Sigma^C, \Sigma^O, I, G)$
**Output:** A STRIPS task $\Pi = (F, O, I, G)$

```
1  q ← I;
2  F ← ∅;                    // Processed facts
3  O ← ∅;                    // Processed operators
4  while ¬(q.empty() ∨ G ⊆ F) ∧ ¬StoppingCondition
   do
5  │   if q.containsFact() then
6  │   │   f ← q.popFact();
7  │   │   F ← F ∪ {f};
8  │   │   for o ∉ O ∧ pre(o) ⊆ F do
9  │   │   │   q.insert(o);
10 │   else
11 │   │   o ← q.popHighPriorityOperator();
12 │   │   O ← O ∪ {o};
13 │   │   for f ∉ F ∧ f ∈ add(o) do
14 │   │   │   q.insert(f);
15 return (F, O, I, G)
```

conditional effects as it is done by Helmert (2009).

Algorithm 1 shows details of our approach. The main difference with respect to the approach by Helmert (2009) is that (1) the algorithm can stop before the queue is empty, and (2) operators are instantiated in a particular order. For these two choice points we suggest an approach that aims at minimizing the size of the partially grounded task, while keeping it solvable. That said, our main focus is the operator ordering, and we only consider a simple stopping condition.

**Stopping condition.** Typical grounding approaches terminate only when the queue is empty, meaning that all (delete-relaxed) reachable facts and operators have been grounded. In partial grounding, we allow the algorithm to stop earlier. Intuitively, this is a good idea because most planning tasks have short plans, usually in the order of at most a few hundred operators, compared to possibly millions of grounded operators. Hence, if the correct operators are selected, partial grounding can potentially stop much sooner than complete grounding. The key issue is how to decide when the probability of finding a plan using the so-far grounded operators is sufficient. Consider the following claims:

1. The grounded task is delete-relaxed solvable iff $G \subseteq F$.

2. The grounded task is solvable iff there exists a plan $\pi$ for $\Pi^{PDDL}$ such that $\pi \subseteq O$.

Item 1 provides a necessary condition for the task to be relaxed-solvable, so grounding should continue at least until $G \subseteq F$. But this is not sufficient, as it does not guarantee that a plan can be found for the non-relaxed task. Item 2 provides an obvious, but difficult to predict, condition for success.

In this work, we consider only a simple stopping condition. To maximize the probability of the task being solvable, it is desirable to ground as many operators as possible. The main constraint on the number of operators to ground are the

resources (time and memory) that can be spent on grounding. For that reason, one may want to continue grounding while these resources are not compromised[1]. We provide a constant $N^{op}$ as a parameter, an estimate on the number of operators that can be grounded given the available resources, and let the algorithm continue as long as $|O| \leq N^{op}$.

If not all actions are grounded, the resulting grounded task is a partial representation of the PDDL input and the overall planning process of grounding and finding a plan for the grounded task is incomplete. We implemented a loop around the overall process that incrementally grounds more actions, when finding the partially grounded task unsolvable. This converges to full grounding, resulting in a complete planner.

**Queue order.** Standard grounding algorithms extract elements from the queue in an arbitrary order – since all operators are grounded, order does not matter. Our algorithm always grounds all facts that have been added to the queue, giving them preference over operators. This ensures that the effects of all grounded operators are part of the grounded task. After all facts in the queue have been processed, our algorithm picks an operator according to a heuristic criterion, which we will call the *priority function*. Some simple priority functions include FIFO, LIFO, or random. Since our aim is to ground all operators of a plan, the priority queue should sort operators by their probability of belonging to a plan. To estimate these probabilities, we use machine learning techniques as detailed in the next section. Additionally, one may want to increase the diversity of selected operators to avoid being misguided by a bias in the estimated probabilities. We consider a simple *round robin* (RR) criterion, which classifies all operators in the queue by the action schema they belong to, and chooses an operator from a different action schema in each iteration. RR works in combination with a priority function that is used to select which instantiation of a given action schema should be grounded next.

We define a *novelty* criterion as a non-trivial priority function that is not based on learning, inspired by novelty pruning techniques that have successfully been applied in classical planning (Lipovetzky and Geffner 2012; 2017). During search, the novelty of a state is defined as the minimum number $m$ for which the state contains a set of facts of size $m$, that is not part of any previously generated state. This can be used to prune states with a novelty $< k$.

We adapt the definition of novelty to operators in the grounding process as follows. Let $\Sigma$ be the set of objects, $a[X]$ an action schema, and $O$ the set of already grounded operators corresponding to all instantiations of $a[X]$. Let $\sigma = \{(x_1, \sigma_1), \ldots, (x_k, \sigma_k)\}$ be an assignment of objects in $\Sigma$ to parameters $X$ instantiating an operator $o$, such that $o \notin O$. Then, the novelty of $o$ is defined as the number of assignments $(x_i, \sigma_i)$ such that there does not exist an operator $o' \in O$ where $x_i$ got assigned $\sigma_i$. In the grounding we will prioritize operators with a higher novelty, which are likely to generate facts that have not been grounded yet.

---

[1]While search can benefit from grounding less operators, an orthogonal pruning method that uses full information of the grounded task, can be employed at that stage (e. g. Heusner et al. (2014)).

## Learning Operator Priority Functions

To guide the grounding process towards operators that are relevant to solve the task, we use a priority queue that gives preference to more promising operators. We use a priority function $f : O \rightarrow [0, 1]$ that estimates whether operators are useful or not. Ideally, we want to assign 1 to operators in an optimal plan and 0 to the rest, so that the number of grounded operators is minimal. We approximate this by assigning to each operator a number between 0 and 1 that estimates the probability that the operator belongs to an optimal plan for the task. This is challenging, however, due to lack of knowledge about the fully grounded task.

We use a learning approach, training a model on small instances of a domain and using it to guide grounding in larger instances. Our training instances need to be small enough to compute the set of operators that belong to any optimal plan for the task. We do this by solving the tasks with a symbolic bidirectional breadth-first search (Torralba et al. 2017) and extracting all operators that belong to an optimal solution.

Before grounding, the only information that we have available is the lifted task $\Pi^{PDDL} = (\mathcal{P}, \mathcal{A}, \Sigma^C, \Sigma^O, I, G)$. Our training data uses this information, consisting of tuples $(I, G, \Sigma^O, o, \{0, 1\})$ for each operator $o$ in a training instance, where $o$ is assigned a value of 1 if it belongs to an optimal solution and 0 otherwise. We formulate our priority functions as a classification task, where we want to order the operators according to our confidence that they belong to the 1 class. To learn a model from this data, we need to characterize the tuple $(I, G, \Sigma^O, o)$ with a set of features. Since training and testing problems have different objects, these features cannot refer to specific objects in $\Sigma^O$, so learning has to be done at the lifted level. We propose relational rules that connect the objects that have instantiated the action schema to the training sample $(I, G, \Sigma^O)$ to capture meaningful properties of an operator. Because different action schemas have different (numbers of) arguments, the features that characterize them will necessarily be different. Therefore, we train a separate model for each action schema $a[X] \in \mathcal{A}$. All these models, however, predict the probability of an operator being in an optimal plan, so the values from two different models are still comparable.

We considered two approaches to conduct the learning: inductive relational trees and classification/regression with relational features.

**Inductive Relational Learning Trees.** Inductive Logic Programming (ILP) (Muggleton and Raedt 1994) is a well-known machine learning approach suitable when the training instances are described in relational logic. ILP has been used, e.g., to learn domain control knowledge for planning (de la Rosa et al. 2011; Krajnansky et al. 2014). We use the Aleph tool (Srinivasan 1999) to learn a tree where each inner node is a predicate connecting the parameters of a to-be-grounded operator to the facts in the initial state or goal, to objects referred to in a higher node in the tree, or to a constant. The nodes are evaluated by checking if there exists a predicate instantiated with the given objects in the initial state or goal. A wildcard symbol ("_") indicates that we do

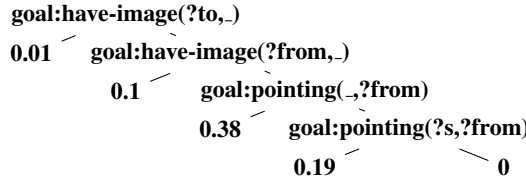

Figure 1: The relational tree learned for the action schema *turn-to(?s - satellite, ?from - direction, ?to - direction)* in the Satellite domain.

not require a particular object, but that any object instantiating the predicate at this position is fine. In Figure 1, the left child corresponds to this check evaluating to false and the right child to true. For a given action, the tree is evaluated by checking if there exists an assignment to the free variables in a path from the root to a leaf node, such that all nodes on the path evaluate to the correct truth value. We then take the real value in the leaf node as an estimate of the probability that the operator is part of an optimal plan. This evaluation is akin to a CSP problem, so we need to keep the depth of the trees at bay to have an efficient priority function.

Figure 1 shows the tree learned for the *turn-to* action schema in Satellite. In this domain, the goal is to take pictures in different modes. Several satellites are available, each with several instruments that support some of the modes. The actions are switching the instruments on and off, calibrating them, turning the satellite into different directions, and taking images. The *turn-to* action changes the direction satellite ?s is looking at. In this case, the learned tree considers that the operators turning to and from relevant directions are more likely part of an optimal plan than turning away from the goal direction. More concretely, if for a to-be-instantiated operator *turn-to(s, from, to)* with objects *s*, *from*, and *to*, there is no goal *have-image(to, _)*, i. e., taking an image in direction *to* using any mode, then the operator is deemed not useful by the trained model, it only has a probability of $1\%$ of belonging to an optimal plan. In the opposite case, and if there is a *have-image* goal in the *from* direction, but no goal *pointing(_, from)*, then the operator is expected to be most useful, with a probability of $38\%$ of being in an optimal plan. This is relevant information to predict the usefulness of *turn-to*. However, there is some margin of improvement since the initial state is ignored.

**Classification and Regression with Relational Features.** An alternative is to use relational rules as features for standard classification and regression algorithms (Kramer, Lavrač, and Flach 2001). Our features are relational rules where the head is an action schema, and the body consists of a set of goal or initial-state predicates, partially instantiated with the arguments of the action schema in the head of the rule, constant objects in $\Sigma^C$, or free variables used in other predicates in the rule. This is very similar to a path from root to leaf in the aforementioned relational trees.

We generate rules by considering all possible predicates and parameter instantiations with two restrictions. First, to guarantee that the rule takes different values for different instantiations of the action schema, one of the arguments in the first predicate in the body of the rule must be bound to a parameter of the action schema. Second, at least one argument of each predicate after the first one, must be bound to a free variable used in a previously used predicate. This aims at reducing the number of features by avoiding redundant rules that can be described as a conjunction of simpler rules. We assume that, if the conjunction of two rules is relevant for the classification task, the machine learning algorithms will be able to infer this.

Most of the generated rules do not provide useful information to predict whether an operator will be part of an optimal plan or not. This is because we brute-force generate all possible rules, including many that do not capture any useful properties. Therefore, it is important to select a subset of relevant features. We do this filtering in two steps. First, we remove all rules that evaluate to the same value in all training instances (e.g., rules that contain `goal:predicate` in the body will never evaluate to true if `predicate` is never part of the goal description in that domain). Then, we use attribute selection techniques in order to filter out those features that are not helpful to predict whether the operator is part of an optimal plan. As an example, the most relevant rule generated for the *turn-to* schema is:

```
turn-to(?s, ?to, ?from) :- goal:have-image(?to, ?M1),
    goal:have-image(?from,?M2), ini:on-board(?I, ?s),
    ini:supports(?I, ?M1), ini:supports(?I, ?M2).
```

This can be read as: "do we have to take images in directions *?to* and *?from* in modes that are supported by one of the instruments on board?". This rule surprisingly accurately describes a scenario where *turn-to* is relevant (and can be complemented with other rules to capture different cases).

Given a planning task and an operator, a rule is evaluated by replacing the arguments in the head of the rule by the objects that are used to instantiate the operator and checking if there exists an assignment to the free variables such that the corresponding facts are present in the initial state and goal of the task. Doing so, we generate a feature vector for each grounded action from the training instances with a binary feature for every rule indicating whether the rule evaluates to true for that operator or not. This results in a training set where for each operator we get a vector of boolean features (one feature per rule), together with a to-be-predicted class that is 1 if the operator is part of an optimal plan for the task, and 0 if not. On this training set, we can use either classification or regression methods to map each operator to a real number. With classification methods we use the confidence that the model has in the operator belonging to the positive class. In regression, the model directly tries to minimize the error by assigning values to 1 for operators in an optimal plan and 0 to others. It is important to note that it is possible that there are two training examples with the same feature vector, but with different values in the target. In these cases, we merge all training examples with the same feature vector and replace them with a single one that belongs to the 1 class if any of the examples did[2].

---

[2]For regression algorithms, we also considered taking the aver-

During grounding, for every operator that is inserted in the queue, we evaluate all rules and call the model to get its priority estimate. To speed-up rule evaluation, we precompute, before grounding, all possible assignments to the arguments of the action schema that satisfy the rule. The computational cost of doing this is exponential in the number of free variables but it was typically negligible for the rules used by our models. We evaluate the relational trees in a similar way.

## Experiments

For the evaluation of our partial grounding approach, we adapted the implementation of the "translator" component of the Fast Downward planning system (FD) (Helmert 2006). The translator parses the given input PDDL files and outputs a fully grounded task in finite-domain representation (FDR) (Bäckström and Nebel 1995; Helmert 2009) that corresponds to the PDDL input. Our changes are minimally invasive, only changing the ordering in which actions are handled and the termination condition, as indicated in Algorithm 1. Therefore, none of the changes affect the correctness of the translator, i. e., the generated grounded planning task will always be a proper FDR task. The changes do not affect the performance too much either, except when using a computationally expensive priority function.

**Experimental Setup.** For the evaluation of our technique, we require domains for which (1) instance generators are available to generate a set of diverse instances small enough for training, and (2) the size of the grounded instances grows at least cubically with respect to the parameters of the generator so that we have large instances that are hard to fully ground, for evaluation. We picked four domains that were part of the learning track of the international planning competition (IPC) 2011 (Blocksworld, Depots, Satellite, and TPP), as well as two domains of the deterministic track of IPC'18 (Agricola and Caldera). For all domains, we used the deterministic track IPC instances and a set of 25 large instances that we generated ourselves for the experiments.

For the training of the models, we used between 40 and 250 small instances, to get enough training data for each action schema. Since the number of grounded actions per schema varies significantly across domains, we individually adapted the number of training instances.

To generate the large instances, we started at roughly the same size as the largest IPC instances, scaling the parameters of the generator linearly when going beyond that. As an example, in Satellite, the biggest IPC instance has around 10 satellites and 20 instruments, which is the size of our smallest instances. In the largest instances that we generated, there are up to 15 satellites and 60 instruments. In Blocksworld, where IPC instances only scale up to 17 blocks, we scale in a different way, starting at 75 blocks and going up to 100, which can still easily be solved by our techniques.

Regarding the domains, we used the typed domain encoding of Satellite from the learning track, which simplifies rule generation, but does not semantically change the domain. In Blocksworld, we use the "no-arm" encoding, which

_______________

age but this resulted in slightly worse results in most cases.

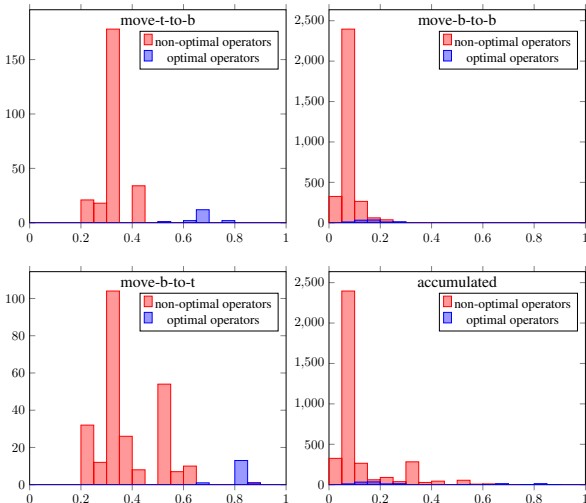

Figure 2: Evaluation of logistic regression in Blocksworld.

shows a cubic explosion in the grounding, in contrast to the "arm" encoding, where the size of the grounded task is only quadratic in the PDDL description.

Beside the static queue orderings, FIFO, LIFO, and novelty-based methods, we experiment with learning-based approaches using classification and regression models. While the former exemplify what is possible without learning, the latter methods aim at grounding only those actions that belong to a plan for a given task. In all cases, we combine the methods with the round robin queue setup (**RR**).

**Learning Framework.** We report results for a *logistic regression classifier* (**LOGR**), *kernel ridge regression* (**KRN**), *linear regression* (**LINR**), and a *support vector machine regressor* (**SVR**). While LINR and LOGR learn linear functions to combine the features, differing mostly in the loss function and underlying model that is used, KRN and SVR are capable of learning non-linear functions. We expect non-linear functions to be useful to combine features, which cannot be done with linear functions. We also report the results of the decision trees learned by **Aleph**. To implement the machine learning algorithms, we use the `scikit` Python package (Pedregosa et al. 2011), which nicely connects to the translator in FD.

For feature selection, i. e., to select which rules are useful to predict the probability of an operator to be in a plan, we used the properties included in the trained models. For each feature (rule) contained in the feature vector, the model returns a weight according to its relevance to discriminate the target vector. After experimenting with multiple different models, we decided to use a decision tree regressor to predict rule usefulness, for all trained models.

We evaluated our models in isolation on a set of validation instances that are distinct from both our training and testing set, and small enough to compute the set of operators that are part of any optimal plan. Figure 2 shows the outcome of the priority function learned by LOGR in Blocksworld. The

| Domain | # | Base | \*Solved within 30min overall\* FIFO | LIFO | RND | Novelty | RR | LINR | RR | LOGR | RR | KRN | RR | SVR | RR | Aleph | RR | \*Last iteration solved within 30min\* FIFO | LIFO | RND | Novelty | RR | LINR | RR | LOGR | RR | KRN | RR | SVR | RR | Aleph | RR |
|---|---|---|---|---|---|---|---|---|---|---|---|---|---|---|---|---|---|---|---|---|---|---|---|---|---|---|---|---|---|---|---|---|
| Agricola-IPC | 20 | **10** | 1 | 1 | 2 | 1 | 3 | 1 | 7 | 5 | 5 | 8 | 5 | 4 | **10** | **10** | 3 | 9 | 9 | 9 | 7 | 10 | 9 | **12** | 11 | **12** | 9 | **12** | 10 | **12** | 11 | 10 |
| Agricola-large | 25 | 4 | 0 | 0 | 0 | 0 | 0 | 0 | 1 | 2 | 1 | 0 | 1 | 0 | **22** | 17 | 0 | 4 | 4 | 4 | 0 | 10 | 1 | **24** | 20 | 23 | 6 | **24** | 19 | **24** | **24** | **24** |
| Blocksworld-IPC | 35 | **35** | 35 | 35 | 35 | 35 | 35 | 35 | 35 | 35 | 35 | 35 | 35 | 35 | 35 | 35 | 35 | 35 | 35 | 35 | 35 | 35 | 35 | 35 | 35 | 35 | 35 | 35 | 35 | 35 | 35 | 35 |
| Blocksworld-large | 25 | 0 | 0 | 0 | 0 | 21 | **25** | 14 | **25** | **25** | 23 | **25** | 24 | **25** | 22 | **25** | **25** | 0 | 0 | 0 | 21 | **25** | 14 | **25** | **25** | 24 | **25** | **25** | **25** | 24 | **25** | **25** |
| Caldera-IPC | 20 | 13 | 13 | 12 | 13 | 9 | 14 | 17 | 18 | 18 | 18 | 11 | 18 | **19** | 18 | 13 | 14 | 17 | 13 | 17 | 15 | 17 | 19 | 19 | 19 | 19 | 17 | 19 | **20** | 19 | 15 | 18 |
| Caldera-large | 25 | 0 | 10 | 0 | 3 | 0 | 5 | 22 | 18 | 18 | **23** | 1 | 19 | 20 | 17 | 0 | 7 | 19 | 0 | 5 | 12 | 16 | **25** | **25** | 24 | **25** | 8 | **25** | **25** | **25** | 0 | 19 |
| Depots-IPC | 22 | 20 | 19 | 20 | 19 | 19 | 20 | 20 | 20 | 19 | **21** | 19 | 20 | 20 | **21** | 20 | 20 | 19 | 20 | 20 | 19 | 20 | 20 | 20 | 19 | **21** | 19 | 20 | 20 | **21** | 20 | 20 |
| Depots-large | 25 | 1 | 0 | 0 | 0 | 0 | 0 | **5** | 3 | 1 | 2 | 2 | 3 | 1 | 4 | 2 | 0 | 0 | 0 | 0 | 0 | 0 | **5** | 3 | 1 | 2 | 2 | 3 | 1 | 4 | 2 | 0 |
| Satellite-IPC | 36 | **36** | 35 | **36** | **36** | 35 | 26 | **36** | 35 | **36** | 35 | 35 | 35 | **36** | 35 | **36** | **36** | 35 | **36** | **36** | **36** | 35 | **36** | **36** | **36** | **36** | **36** | **36** | **36** | **36** | **36** | **36** |
| Satellite-large | 25 | 0 | 0 | 0 | 0 | 1 | 0 | 0 | 11 | 0 | 14 | **15** | 14 | 0 | 14 | 1 | 1 | 0 | 0 | 0 | 1 | 0 | 0 | 14 | 0 | 16 | **19** | 15 | 0 | 16 | 1 | 1 |
| TPP-IPC | 30 | **30** | 30 | 30 | 30 | 30 | 30 | 26 | 28 | 30 | 30 | 30 | 30 | 30 | 29 | 30 | 30 | 30 | 30 | 30 | 30 | 30 | 30 | 30 | 30 | 30 | 30 | 30 | 30 | 30 | 30 | 30 |
| TPP-large | 25 | 7 | 5 | **8** | 2 | 6 | **8** | 1 | 2 | 4 | 4 | 5 | 6 | 4 | 6 | **8** | 6 | 6 | **8** | 6 | 6 | **8** | 5 | 5 | 7 | 5 | 5 | 6 | 7 | **9** | **8** | 6 |
| $\sum$ | 313 | 156 | 148 | 142 | 140 | 157 | 166 | 177 | 203 | 193 | 211 | 186 | 210 | 194 | **233** | 197 | 177 | 174 | 155 | 161 | 182 | 206 | 199 | 248 | 227 | 248 | 211 | 250 | 228 | **255** | 207 | 224 |

Table 1: Number of instances solved by the baseline with full grounding (Base), and incremental grounding with static action orderings (FIFO, LIFO, random (RND)), novelty-based ordering, and several learning models (see text). "RR" indicates that we use a separate priority queue for each action schema, taking turns over the schemas. Best coverage highlighted in **bold** face.

bars indicate the number of operators across all validation instances that got a priority in a given interval, highlighting operators from optimal plans in a different color. The plots nicely illustrate that the priority function works very well for the action schemas *move-t-to-b* and *move-b-to-t*, where it is able to distinguish "optimal" from "non-optimal" operators. The distinction works not so well for *move-b-to-b*, but in general gives a significantly lower priority to this action schema. Another important observation is that the total number of grounded *move-b-to-b* actions is much higher than that of the other two action schemas.

Projecting these observations to the grounding process, we expect the model to work well when used in a single priority queue, since it will prioritize *move-t-to-b* and *move-b-to-t* (which are the only ones needed to solve any Blocksworld instance) over *move-b-to-b* (which is only needed to optimally solve a task). On the validation set, grounding all operators with a priority of roughly $> 0.6$ suffices to solve the tasks, pruning all *move-b-to-b* operators and most non-optimal ones of the other schemas. RR in contrast will ground an equal number of all action schemas, including many unnecessary operators. These conjectures are well-supported by the plots in Figure 3.

When working with machine learning techniques, there is always the risk of overfitting. In our case the results on the training set are very similar to those on the validation set shown in Figure 2, suggesting that overfitting is not an issue in our setup. The results in other domains are similar.

**Incremental Grounding.** We use the incremental approach, where the first iteration grounds a given task until the goal is found to be relaxed-reachable. The left half of Figure 3 shows detailed information on how many operators need to be grounded until this is achieved for different priority functions. We discuss details later. In case this first iteration fails, i.e., the partial task is not solvable, we set a minimum number of operators to be grounded in the next iteration by using an increment of 10 000 operators. This

strategy does not aim to maximize coverage but rather to find out what is the minimum number of operators that need to be grounded to solve a task for each priority function (with a granularity of 10 000 operators). The number of operators that were necessary to actually solve a given instance is illustrated in the right half of Figure 3.

For all configurations, after grounding, we run the first iteration of the LAMA planner (Richter and Westphal 2010), a good standard configuration for satisficing planning that is well integrated in FD. We also use LAMA's first iteration as a baseline on a fully grounded task, with runtime and memory limits for the entire process of 30 minutes and 4GB. All other methods perform incremental grounding using their respective priority function. We allowed for a total of 5 hours and 4GB for the incremental grounding, while restricting the search part to only 10 minutes per iteration to keep the overall runtime of the experiments manageable.

We show coverage, i.e., number of instances solved, in Table 1, with the time and memory limits mentioned in the previous subsection. The left part of the table considers instances as solved when the overall incremental grounding process (including finding a plan) finished within 30 min. In the right part, we approximate the results that could be achieved with a perfect stopping condition by considering an instance as solved if the last iteration, i.e., the successful grounding and search, finished within 30 min.

The baseline (**Base**) can still fully ground most instances except in Caldera and TPP, but fails to solve most of the large instances with up to 9 million operators. We scaled instances in this way so that a comparison of the number of grounded operators to the baseline is possible; further scaling would make full grounding impossible.

The table nicely shows that the incremental grounding approach, where several iterations of partial grounding and search are performed (remember that we only allow 10min for the search), significantly outperforms the baseline, even when considering an overall time limit of 30min. In fact, all instances in Blocksworld can be solved in less than 10s

by LOGR. This illustrates the power of our approach when the learned model captures the important features of a domain. The static orderings typically perform worse than the baseline, only the novelty-based ordering can solve more instances in Blocksworld, and in Caldera when using RR.

The plots in Figure 3 shed further light on the number of operators when (leftmost two columns) the goal is relaxed reachable in the first iteration and (rightmost two columns) the number of operators needed to actually solve the task. Each data point corresponds to a planning instance, with the number of ground actions of a fully grounded task on the x-axis. The y-axis shows the number of grounded actions for several priority functions, including FIFO (LIFO in TPP), novelty, the learned model that has the highest reduction on the number of grounded actions, and Aleph.

In general, the models capture the features of most domains quite accurately, leading to a substantial reduction in the size of the grounded task, and still being able to find a solution. The plots show that our models obtain a very strong reduction of the number of operators in the partially grounded task in Agricola, Blocksworld, and Caldera; some reduction (one order of magnitude) in Depots, and Satellite, and a small reduction in TPP. In terms of the size of the partially grounded tasks, different learning models perform best in different domains, and there is not a clear winner. In comparison, the baselines FIFO, LIFO, and Random do not significantly reduce the size of the grounded task in most cases, with a few exceptions like LIFO in TPP and FIFO in Caldera. The novelty criterion is often the best method among those without learning.

Grounding a delete-relaxed reachable task with less operators is often beneficial, but may be detrimental for the coverage if the task is unsolvable, as happens for the Novelty method in Agricola or the LIFO method in TPP. This also explains why the learning models with highest reductions in some domains (e.g. LOGR in Agricola) are not always the same as the ones with highest coverage. The RR queue mechanism often grounds more operators before reaching the delete-relaxed goal but this makes the first iteration solvable more often leading to more stable results. The exception is Aleph, where RR has the opposite effect, making the partially grounded tasks unsolvable.

## Related Work

Some approaches in the literature try to alleviate the grounding problem, e. g. by avoiding grounding facts and operators unreachable from the initial state (Helmert 2009), reformulating the PDDL description by splitting action schemas with many parameters (Areces et al. 2014), or using symmetries to avoid redundant work during the grounding process (Röger, Sievers, and Katz 2018).

Lifted planning approaches that skip grounding entirely (Penberthy and Weld 1992) have lost popularity due to the advantages of grounding to speed-up the search and allow for more informative heuristics which are not easy to compute in a lifted level. Ridder and Fox (2014) adapted the delete-relaxation heuristic (Hoffmann and Nebel 2001a) to the lifted level. This is related to our partial grounding approach since their relaxed plan extraction mechanism can

be used to obtain a grounded task where the goal is relaxed reachable, and it could be used to enhance the novelty and learning priority functions that we use here.

There are many approaches to eliminate irrelevant facts and operators from grounded tasks (Nebel, Dimopoulos, and Koehler 1997; Hoffmann and Nebel 2001b; Haslum, Helmert, and Jonsson 2013; Torralba and Kissmann 2015). The closest to our approach is under-approximation refinement (Heusner et al. 2014), which also performs search with a subset of operators. However, all these techniques use information from the fully grounded representation to decide on the subset of relevant operators, so are not directly applicable in our setting. The results of our learning models (see Figure 2) show that applying learning to identify irrelevant operators is a promising avenue for future research.

Recently, Toyer et al. (2018) introduced a machine learning approach to learn heuristic functions for specific domains. This is similar to our work, in the sense that a heuristic estimate has been learned, though for states in the search, not actions in the grounding. Furthermore, the authors used neural networks instead of our, more classical, models.

## Conclusion

In this paper, we proposed an approach to partial grounding of planning tasks, to deal with tasks that cannot be fully grounded under the available time and memory resources. Our algorithm heuristically guides the grounding process giving preference to operators that are deemed most relevant for solving the task. To determine which operators are relevant, we train different machine learning models using optimal plans from small instances of the same domain. We consider two approaches, a direct application of relational decision trees, and using relational features with standard classification and regression algorithms. The empirical results show the effectiveness of the approach. In most domains, the learned models are able to identify which operators are relevant with high accuracy, helping to reduce the number of grounded operators by several orders of magnitude, and greatly increasing coverage in large instances.

**Acknowledgments** This work was supported by the bilateral project of the German Academic Exchange Service (DAAD) and the Argentinian Ministry of Science, Technology, and Productive Innovation (MinCyT) number DA/16/01 "Optimizing Planning Domains". Daniel Gnad was partially supported by the German Research Foundation (DFG), under grant Nr. HO 2169/6-1, "Star-Topology Decoupled State Space Search".

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

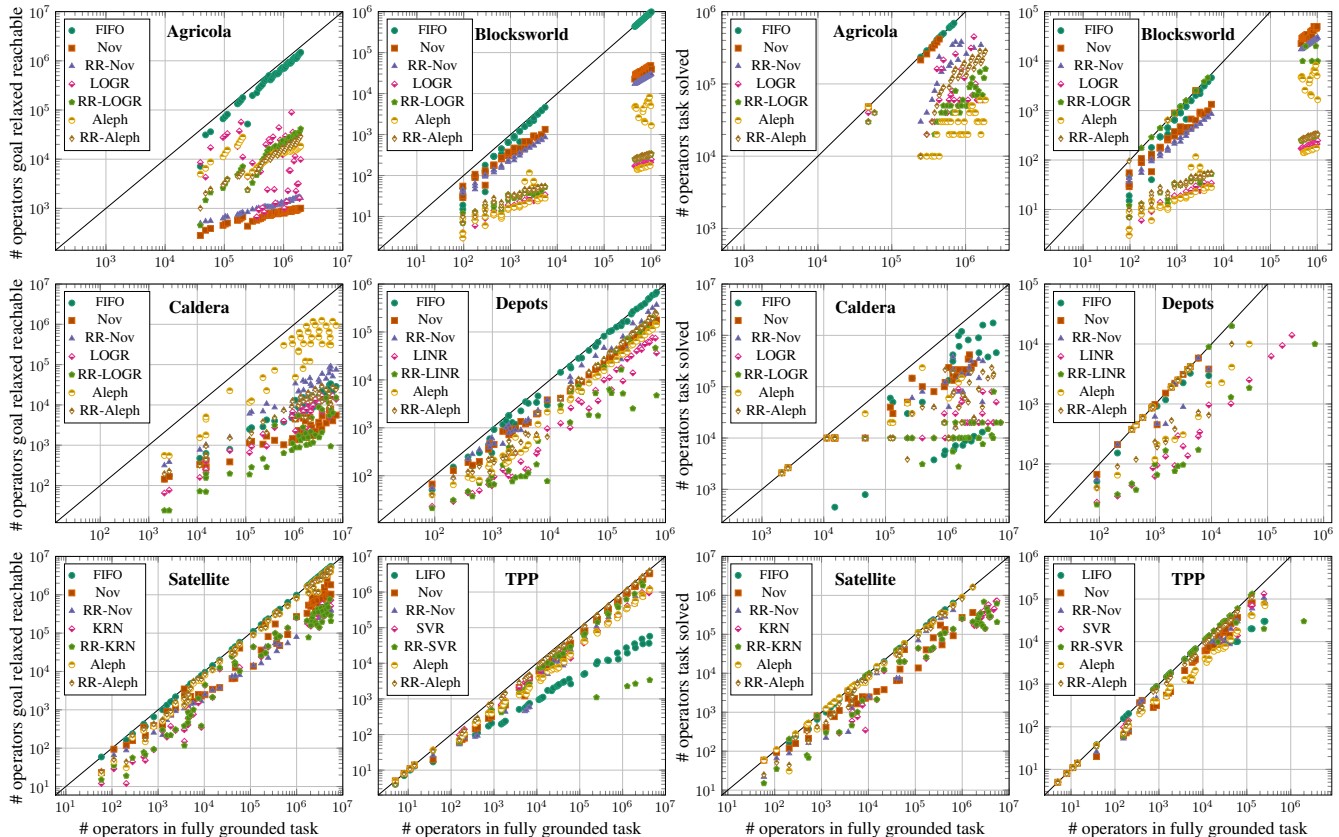

Figure 3: The scatter plots show the number of operators of a fully grounded task on the x-axis. The y-axis shows the number of operators that are needed to make the goal reachable in the grounding (leftmost two columns), and the number of operators that are needed to solve the task (rightmost two columns), for several priority functions.

de la Rosa, T.; Celorrio, S. J.; Fuentetaja, R.; and Borrajo, D. 2011. Scaling up heuristic planning with relational decision trees. *JAIR* 40:767–813.

Fikes, R. E., and Nilsson, N. 1971. STRIPS: A new approach to the application of theorem proving to problem solving. *Artificial Intelligence* 2:189–208.

Haslum, P.; Helmert, M.; and Jonsson, A. 2013. Safe, strong, and tractable relevance analysis for planning. In *Proc. ICAPS'13*.

Helmert, M. 2006. The Fast Downward planning system. *JAIR* 26:191–246.

Helmert, M. 2009. Concise finite-domain representations for PDDL planning tasks. *Artificial Intelligence* 173:503–535.

Heusner, M.; Wehrle, M.; Pommerening, F.; and Helmert, M. 2014. Under-approximation refinement for classical planning. In *Proc. ICAPS'14*.

Hoffmann, J., and Nebel, B. 2001a. The FF planning system: Fast plan generation through heuristic search. *JAIR* 14:253–302.

Hoffmann, J., and Nebel, B. 2001b. RIFO revisited: Detecting relaxed irrelevance. In *Proc. of ECP'01*, 325–336.

Krajnansky, M.; Buffet, O.; Hoffmann, J.; and Fern, A. 2014. Learning pruning rules for heuristic search planning. In *Proc. of ECAI'14*, 483–488.

Kramer, S.; Lavrač, N.; and Flach, P. 2001. Propositionalization

approaches to relational data mining. In *Relational data mining*. Springer. 262–291.

Lang, T., and Toussaint, M. 2009. Relevance grounding for planning in relational domains. In *Proc. of ECML'09*, 736–751.

Lipovetzky, N., and Geffner, H. 2012. Width and serialization of classical planning problems. In *Proc. of ECAI'12*, 540–545.

Lipovetzky, N., and Geffner, H. 2017. A polynomial planning algorithm that beats LAMA and FF. In *Proc. ICAPS'17*, 195–199.

McDermott, D.; Ghallab, M.; Howe, A.; Knoblock, C.; Ram, A.; Veloso, M.; Weld, D.; and Wilkins, D. 1998. *The PDDL Planning Domain Definition Language*. The AIPS-98 Planning Competition Comitee.

Miller, D.; Alford, R.; Applebaum, A.; Foster, H.; Little, C.; and Strom, B. 2018. Automated adversary emulation: A case for planning and acting with unknowns.

Muggleton, S., and Raedt, L. D. 1994. Inductive logic programming: Theory and methods. *JLP* 19/20:629–679.

Nebel, B.; Dimopoulos, Y.; and Koehler, J. 1997. Ignoring irrelevant facts and operators in plan generation. In *Proc. of ECP'97*, 338–350.

Pednault, E. P. 1989. ADL: Exploring the middle ground between STRIPS and the situation calculus. In *Proc. of KR'89*, 324–331.

Pedregosa, F.; Varoquaux, G.; Gramfort, A.; Michel, V.; Thirion, B.; Grisel, O.; Blondel, M.; Prettenhofer, P.; Weiss, R.; Dubourg,

V.; Vanderplas, J.; Passos, A.; Cournapeau, D.; Brucher, M.; Perrot, M.; and Duchesnay, E. 2011. Scikit-learn: Machine learning in Python. *Journal of Machine Learning Research* 12:2825–2830.

Penberthy, J. S., and Weld, D. S. 1992. UCPOP: A sound, complete, partial order planner for ADL. In Nebel, B.; Swartout, W.; and Rich, C., eds., *Principles of Knowledge Representation and Reasoning: Proc. of the 3rd International Conference (KR-92)*, 103–114. Cambridge, MA: Morgan Kaufmann.

Richter, S., and Westphal, M. 2010. The LAMA planner: Guiding cost-based anytime planning with landmarks. *JAIR* 39:127–177.

Richter, S.; Westphal, M.; and Helmert, M. 2011. LAMA 2008 and 2011 (planner abstract). In *IPC 2011 planner abstracts*, 50–54.

Ridder, B., and Fox, M. 2014. Heuristic evaluation based on lifted relaxed planning graphs. In *Proc. ICAPS'14*, 244–252.

Röger, G.; Sievers, S.; and Katz, M. 2018. Symmetry-based task reduction for relaxed reachability analysis. In *Proc. of ICAPS'18*, 208–217.

Srinivasan, A. 1999. The Aleph manual.

Torralba, Á., and Kissmann, P. 2015. Focusing on what really matters: Irrelevance pruning in merge-and-shrink. In *Proc. of SOCS'15*, 122–130.

Torralba, Á.; Alcázar, V.; Kissmann, P.; and Edelkamp, S. 2017. Efficient symbolic search for cost-optimal planning. *Artificial Intelligence* 242:52–79.

Toyer, S.; Trevizan, F.; Thiebaux, S.; and Xie, L. 2018. Action schema networks: Generalised policies with deep learning. In McIlraith, S., and Weinberger, K., eds., *Proc. of the 32nd AAAI Conference on Artificial Intelligence (AAAI'18)*. AAAI Press.

Yoon, S. W.; Fern, A.; and Givan, R. 2008. Learning control knowledge for forward search planning. *Journal of Machine Learning Research* 9:683–718.
