# OpenReview forum: "Learning How to Ground a Plan - Partial Grounding in Classical Planning"
_icaps-conference.org/ICAPS/2019/Workshop/HSDIP_

### Official Review · AnonReviewer1 · 2019-04-05
**This paper presents highly relevant work to the field of classical planning and I do not see a reason to not accept this work at the workshop**

**Rating:** 9
**Confidence:** 4

**Review:**

Short summary of the paper:
This works introduces partial grounding techniques for planning tasks, shows how machine learning techniques can be used to prioritize the operator order of the grounding process, and presents an empirical evaluation of the techniques on
multiple planning domains.

Detailed review:
In planning, most planners nowadays perform grounding as a preprocess step before search. Therefore even the strongest search algorithms won't be of use if the planner is not even able to complete the grounding step. This was indeed the
case in some problems of the latest international planning competition (IPC 19), therefore this work investigates a very relevant field for planning and fits in the scope of the workshop.
The paper reads very well and does a good job in presenting the problem and the idea of partial grounding and operator ordering. Related work is cited, although it could relate itself to the recent introduction of action schema networks
(Toyer et al., 2018), which also apply machine learning techniques to the lifted task representation, although these are used to guide search. Nevertheless, the presented techniques are novel and for the most part clearly defined. An extensive empirical evaluation shows how the different techniques compare to each other and show that the partial grounding techniques can significantly increase the coverage of a planner, although there is not one dominating technique.
All in all, this paper presents highly relevant work to the field of classical planning and I do not see a reason to not accept this work at the workshop.

My only real criticism is that the presentation of the ILP and the classification approach is somewhat informal and a bit convoluted. While the remainder of the paper is very well and fluently written I had to reread this
section several times before fully understanding the underlying concepts. I think either a more formal setting or a more detailed example on both approaches could be helpful for the reader.

Sam Toyer, Felipe W. Trevizan, Sylvie Thiébaux, Lexing Xie:
Action Schema Networks: Generalised Policies With Deep Learning. AAAI 2018: 6294-6301

Minor comments:

- I would argue that a stopping condition is a condition on when the algorithm stops, but in this work it is a condition on when the algorithm continues
- 'Let N^op be a constant, [...]: require the algorithm to continue while...':  the colon does not really make sense here. Maybe just end the sentence and start with 'We require'.
- The text of Figure 1 is hard to read on printed paper, consider the use of bold font
- Fast Downward is cited twice
- 'Ridder and Fox (2014) (2014)' => duplicate year

---

> ### Author Response · Authors · 2019-04-11
> **Thank you for the helpful comments!**
>
> We will fix the minor issues and try to clarify the part you mentioned. We will also cite the work by Toyer et al., although we think that the connection is somewhat weak.

---

### Official Review · AnonReviewer2 · 2019-04-06
**Very well-written paper tackling an interesting problem**

**Rating:** 9
**Confidence:** 4

**Review:**

This paper presents an approach for only partially grounding classical planning
tasks which are too large to be fully grounded by common grounding algorithms.
The approach uses machine learning techniques to estimate the probability of
operators belonging to a plan of the task, using information from small
instances of the same domain. These operators are then considered first for
grounding, which can be stopped early if otherwise risking to run out of
resources. The resulting partially grounded task is not guaranteed to be
solvable even if the original task was. An experimental evaluation shows that
the approach works well in several IPC domains where very large tasks can be
solved with partial grounding.

Since this paper is already published at AAAI and the submitted version is
identical (and not a longer technical-report style variant), I will refrain
from writing a full review. While the paper does not directly fall into the
category of search or heuristic for planning, it addresses the problem of
solving domains that are challenging (because of their size) as as such also
fits the scope of the workshop. The paper is very well written and easy to
follow and hence my recommendation is a clear accept for the workshop.

If I had to point out anything that could be improved (in case the authors
would actually like to turn this into a one page longer extended paper), then I
would to suggest to include a description of the used machine learning
techniques because many researchers at ICAPS are probably not very familiar
with these. More importantly, I think it would be interesting to discuss why
particular ML methods worked or didn't work for the purpose of the paper, or,
if that cannot be explained easily, at least state what differences were
observed, what parameters ended up being used and how this affected results.

---

> ### Author Response · Authors · 2019-04-11
> **Thank you for the good comments!**
>
> Regarding relevance to the workshop: we somewhat agree that the paper as is does not completely fit into the topic of "search and heuristics". If the grounding process is interpreted as some kind of search - in the "delete-relaxed search space" - then the methods we propose, learning and novelty, aim at guiding this search to ground those actions that are deemed useful to achieve the goals. We also think that more interesting work can be done into this direction, not necessarily using learning, but trying to obtain structural information on the lifted level, similar to the heuristics we are used to when doing search in the grounded task.
>
> Unfortunately, we have the same trouble in understanding why certain ML models are able to capture important aspects of some domains, but not of others, as people generally have when trying to understand why ML models work. There is no easy way of telling why a certain model has chosen a particular rule to be important. We think that the rule generation (for all models but Aleph) is more important. If the ML methods are fed with a set of good rules, then it might even be a matter of being lucky if the method "sees" a connection between rules that actually contain useful information for a domain. That's why we spent quite some effort in filtering the set of (all) rules by (1) eliminating those that do not contain any information, and (2) ranking the remaining rules by a fairly easy to understand model (decision tree) by the "usefulness" this method assigns to them.
>
> We will try to give concise descriptions of the ML methods we used.

---

### Meta-Review · Program_Chairs · 2019-04-25

**Recommendation:** Accept
**Confidence:** 5

**Metareview:**

Dear Authors,
thank you very much for your submission. We are happy to inform you that
we have decided to accept it and we look forward to your talk in the workshop.
Please, go over the feedback in the reviews and correct or update your papers
in time for the camera ready date (May 24).
Best regards
HSDIP organizers